# Assessing the Impact of Lightning NOx Emissions in CMAQ Using Lightning Flash Data from WWLLN over the Contiguous United States

**Daiwen Kang** [1,*] , **Christian Hogrefe** [1], **Golam Sarwar** [1], **James D. East** [1,2], **J. Mike Madden** [1], **Rohit Mathur** [1] and **Barron H. Henderson** [3]

1   Center for Environmental Measurement & Modeling, U.S. Environmental Protection Agency, Durham, NC 27711, USA
2   Department of Civil, Construction, and Environmental Engineering, North Carolina State University, Raleigh, NC 27695, USA
3   Office of Air Quality Planning and Standards, U.S. Environmental Protection Agency, Durham, NC 27711, USA
*   Correspondence: kang.daiwen@epa.gov

**Abstract:** Comparison of lightning flash data from the National Lightning Detection Network (NLDN) and from the World Wide Lightning Location Network (WWLLN) over the contiguous United States (CONUS) for the 2016–2018 period reveals temporally and spatially varying flash rates that would influence lightning $NO_x$ ($LNO_x$) production due to known detection efficiency differences especially during summer months over land (versus over ocean). However, the lightning flash density differences between the two networks show persistent seasonal patterns over geographical regions (e.g., land versus ocean). Since the NLDN data are considered to have higher accuracy (lightning detection with >95% efficiency), we developed scaling factors for the WWLLN flash data based on the ratios of WWLLN to NLDN flash data over time (months of year) and space. In this study, sensitivity simulations using the Community Multiscale Air Quality (CMAQ) model are performed utilizing the original data sets (both NLDN and WWLLN) and the scaled WWLLN flash data for $LNO_x$ production over the CONUS. The model performance of using the different lightning flash datasets for ambient $O_3$ and $NO_x$ mixing ratios that are directly impacted by $LNO_x$ emissions and the wet and dry deposition of oxidized nitrogen species that are indirectly impacted by $LNO_x$ emissions is assessed based on comparisons with ground-based observations, vertical profile measurements, and satellite products. During summer months, the original WWLLN data produced less $LNO_x$ emissions (due to its lower lightning detection efficiency) compared to the NLDN data, which resulted in less improvement in model performance than the simulation using NLDN data as compared to the simulation without any $LNO_x$ emissions. However, the scaled WWLLN data produced $LNO_x$ estimates and model performance comparable with the NLDN data, suggesting that scaled WWLLN may be used as a substitute for the NLDN data to provide $LNO_x$ estimates in air quality models when the NLDN data are not available (e.g., due to prohibitive cost or lack of spatial coverage).

**Keywords:** lightning $NO_x$; WWLLN; air quality; CMAQ; oxidized nitrogen deposition

## 1. Introduction

Lightning produces nitrogen oxides ($NO_x$ = NO [nitric oxide] + $NO_2$ [nitrogen dioxide]) in the mid and upper troposphere and are estimated to contribute 10–15% of the total global $NO_x$ emissions budget [1], and as the only natural emissions source aloft from the Earth's surface, exerts a profound influence on atmospheric chemistry across the troposphere [2–6]. Due to stringent control measures for $NO_x$ emissions from fossil fuel combustion in response to tightened ozone ($O_3$) standards, significant reductions in anthropogenic $NO_x$ emissions have occurred in the past two decades in the United States

(https://www.epa.gov/air-trends/air-quality-national-summary#air-quality-trends (accessed on 4 August 2022)) [7] and many other parts of the world [8]. As a result, lightning $NO_x$ ($LNO_x$) plays an increasingly important role in both atmospheric composition and chemistry. To account for this essential source in atmospheric chemistry models, $LNO_x$ production and distribution were initially implemented through various parameterization schemes in global models [9–14]. When lightning flash data from ground-based networks became readily available, a parameterization scheme for CMAQ model flash rates was derived such that predicted monthly flashes were scaled to the NLDN observed flashes [15]. This scheme was improved such that inline $LNO_x$ production was based on hourly observed lightning flashes and implemented in the Community Multiscale Air Quality (CMAQ) model (version 5.2 and beyond) for retrospective applications [16,17]. The initial applications of this $LNO_x$ production scheme focused on the Contiguous United States (CONUS) and were based on lightning flashes observed by the National Lightning Detection Network (NLDN) which covers the CONUS with very high detection efficiency [18,19]. As the spatial scales of atmospheric modeling have expanded from regional to hemispheric to global scales [20,21] and new lightning datasets have become available, there is a strong need to include more accurate $LNO_x$ emissions in air quality models at these larger scales. The lightning flashes from the World Wide Lightning Location Network (WWLLN, operated by the University of Washington: http://www.wwlln.net (accessed on 4 August 2022)) is a suitable candidate source for lightning flash data due to its global coverage, although its detection efficiency is lower than the >95% detection efficiency of NLDN [22].

The direct impacts of $LNO_x$ emissions on the tropospheric $NO_x$ and $O_3$ mixing ratios have been assessed in many $LNO_x$-related studies previously [3,4,14,17]. The indirect impact of $LNO_x$ emissions on subsequent formation of other nitrogen derivatives and their ultimate deposition primarily in the form of aerosol nitrate ($NO_3^-$) was recognized but traditionally not included in studies of nitrogen deposition [3,23]. A few studies that included and isolated $LNO_x$ impacts using regional models focused on some localized regions and reported significant model underestimate of wet $NO_3^-$ deposition [15,24], and it was found that inclusion of $LNO_x$ production increased the mean wet deposition of nitrate by 43% [15]. Taking advantage of more than 10 years' advancement in model development and lightning detection techniques, as well as the annual sensitivity simulations for different $LNO_x$ configurations, a section of this paper is devoted to assessing the impact of $LNO_x$ emissions on wet and dry $NO_3^-$ deposition over the CONUS.

The research objectives in this study include: (1) compare WWLLN to NLDN in terms of flash rates and location accuracy over the CONUS domain and evaluate their impact on model performance, (2) scale the lightning flash rates in WWLLN by the NLDN to WWLLN flash ratios and assess the effects of scaling factors on model performance, and (3) assess the impact of $LNO_x$ emissions on wet and dry $NO_3^-$ deposition in general, and the sensitivity of modeled wet and dry $NO_3^-$ deposition to different $LNO_x$ configurations in particular. To achieve these research objectives, we apply all inventories over the CONUS domain for the 2016 annual simulations, similar to our previous development and applications using NLDN data over the CONUS domain [5,16,17]. In Section 2, we describe the data and methodologies of the model simulations and their evaluation. Section 3 provides the comparison of lightning flashes from NLDN and WWLLN over the CONUS and the methodology developed to scale the WWLLN data based on the ratios between NLDN and WWLLN. Section 4 presents the analysis and evaluation of the model performance with the different configurations of $LNO_x$ and concluding remarks are presented in Section 5.

## 2. Data and Methodology

### 2.1. Lightning Flash Data

Lightning flash data from two ground-based lightning detection networks were acquired for 2016–2018. The NLDN provides Cloud-to-Ground (CG) lightning observations with a detection efficiency of >95% and a location accuracy of about 150 m [18,19,25] over the CONUS. The WWLLN provides global lightning data with lower detection efficiency

(60–80% for cloud-to-ground strikes, varying between geographic regions) and lower location accuracy (4–5 km) [22,26,27] compared to both NLDN and the satellite-based Lightning Imaging Sensor (LIS) observations [28,29]. Since WWLLN has global coverage, even with its relatively lower detection efficiency and location accuracy compared to NLDN, it has the potential for being a good option to estimate $LNO_x$ emissions for applications beyond the CONUS where NLDN data are not available.

### 2.2. Observations and Analysis Techniques

The EPA's Air Quality System (AQS; https://www.epa.gov/aqs (accessed on 4 August 2022)) datasets were used to assess the impact of $LNO_x$ on model performance of surface $O_3$ for several model simulations using the NLDN and WWLLN datasets. Since $LNO_x$ is produced primarily in the mid-to-upper troposphere, it is important to assess how the different $LNO_x$ configurations affect the vertical profile and column density of related species. For this purpose, ozonesonde measurements over the CONUS and tropospheric $NO_2$ Vertical Column Densities (VCDs) from the Ozone Monitoring Instrument (OMI) operational retrieval products (Level 2 and version 3) [30] released by the NASA Goddard Earth Sciences Data and Information Service Center (GES DISC) were compared to the model output. Data from the National Atmospheric Deposition Program's National Trends Network (NADP/NTN, http://ndp.slh.wisc.edu/ntn (accessed on 4 August 2022)) were used to evaluate the model estimated wet and dry deposition of $NO_3^-$ [31], another important component impacted by the magnitude and variability of $LNO_x$ estimates in chemical transport model simulations.

$LNO_x$ emissions and the impact on air quality exhibit distinct spatial variations [4,5], so analysis was conducted for the entire CONUS domain along with the U.S. NOAA climate regions (https://www.ncei.noaa.gov/monitoring-references/maps/us-climate-regions (accessed on 4 August 2022)) as shown in Figure 1. Also shown in Figure 1 are the ozonesonde measurement locations within the CONUS domain. The Root Mean Square Error (RMSE), Normalized Mean Error (NME), Mean Bias (MB), Normalized Mean Bias (NMB), and Correlation Coefficient (r) commonly used in the air quality modeling community to assess model performance [32] were calculated for the different model configurations using available observations.

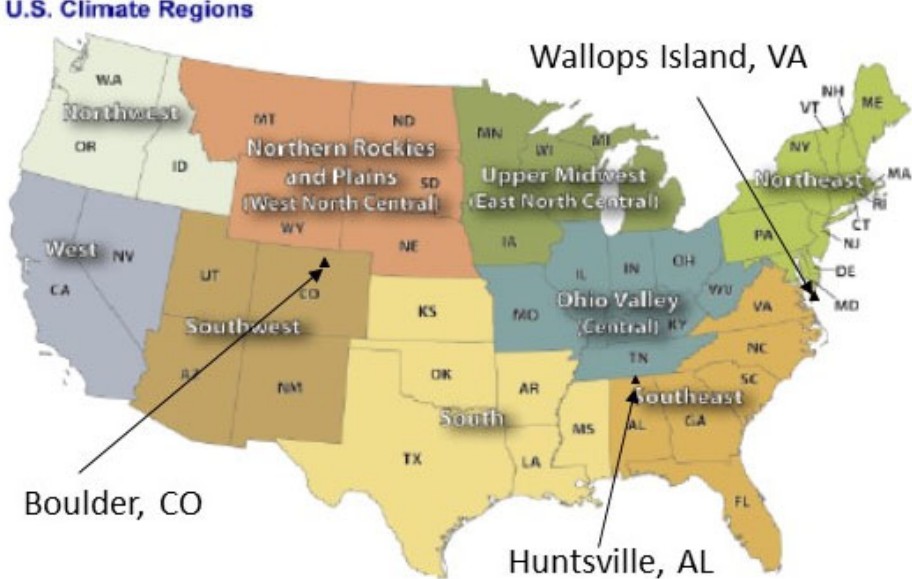

**Figure 1.** U.S. climate regions and locations of ozonesonde measurement.

*2.3. Model Configurations*

The CMAQ model configuration and inputs used in this study are similar to those in [32]. As such, only the key configuration components specific to this study or deviations from [32] are summarized here. This study utilizes CMAQ version 5.3.2 https://doi.org/10.5281/zenodo.4081737 (accessed on 4 August 2022)) with 12 km horizontal grid spacing and 35 vertical layers with varying thickness from the surface to 50 hPa on a Lambert-conformal projection. The Weather Research and Forecasting model version 3.8 (WRFv3.8) with lightning assimilation [33] based on NLDN lightning flash data was used for input meteorology for all the CMAQ simulations. Emissions inputs were generated based on the 2016 National Emissions Inventory (NEI) version 2 platform (https://www.epa.gov/air-emissions-modeling/2016v2-platform (accessed on 4 August 2022)). Lateral boundary conditions for the CONUS domain were provided by a Northern Hemispheric simulation using v5.3 Hemispheric CMAQ (HCMAQ) [20] employing the Carbon Bond 6 chemical mechanism [34]) and the detailed halogen [35,36] and dimethyl sulfide [37] chemistry. Annual simulations were conducted for the entire year of 2016, though the analysis is focused on summertime when the lightning activity over the CONUS is most pronounced. Four simulations to investigate the CMAQ model's sensitivity to different $LNO_x$ configurations were performed: BASE (no $LNO_x$), NDLN ($LNO_x$ generated using hourly NLDN lightning flash data), WWLLN ($LNO_x$ generated using WWLLN lightning flash data), and WWLLNs ($LNO_x$ was generated using scaled WWLLN lightning flash data; described in Section 3). It is important to note that the BASE model does include $LNO_x$ indirectly through the boundary conditions, which were provided by hemispheric model simulations with climatological lightning emissions.

## 3. The Comparison of Lightning Flashes Detected by NLDN and WWLLN and the Adaptation of WWLLN Data for $LNO_x$ Emissions

*3.1. Temporal and Spatial Distributions*

The two ground-based lightning detection networks, NLDN and WWLLN, have employed very different lightning detection technologies and waveform signal processing algorithms to determine the flash intensity and locations [22] that resulted in varying detection efficiency and location accuracy. Specifically, NLDN covers the CONUS using wideband sensors that operate from approximately 400 Hz–400 kHz radio wave detection and therefore is an extremely accurate lightning detection network and is often considered as the "ground truth" for lightning observations. Conversely, the WWLLN has global coverage using Very Low Frequency (VLF, 3–30 kHz) radio wave detection (VLF waves propagate through the ionosphere with relatively low attenuation, enabling the detection of these radio atmospherics at great distances from the lightning discharge) [22]. Before assessing the impact of $LNO_x$ production on air quality using lightning flashes from these two networks, we first examine how the detected lightning flash rates from these two networks compare in time and space. The NLDN lightning data provides CG flashes and cloud-to-cloud (CC) flashes separately with the CG flashes considered to be more accurately detected. For this reason, the CG flashes are directly used to generate $LNO_x$ in CMAQ, whereas the CC flashes are calculated based on climatological CG/CC ratios [16]. The WWLLN dataset provides total lightning flashes (no distinction between CG and CC flashes, but mostly CG flashes). To account for the uneven spatial distribution of WWLLN detection efficiency over the globe, adjust the WWLLN lightning density, and ensure that the WWLLN lightning activities in different places are comparable [38], global time-varying relative detection efficiency maps (DEmaps; scaling factors) are also available (http://wwlln.net/deMaps (accessed on 4 August 2022)). The CG flashes from NLDN and the total lightning flashes from WWLLN corrected by the values in DEmaps are employed in the comparisons of NLDN and WWLLN lightning flashes and lightning flash rates presented here (the "original" or "raw" WWLLN data as referred in the paper is corrected by the DEmaps).

Figure 2a presents the monthly total lightning flashes over the CONUS during 2016 through 2018 for both the NLDN and WWLLN lightning data. For all three years, the NLDN network detected more lightning flashes than the detection-efficiency corrected WWLLN network during the summer months (June, July, and August), but similar or slightly fewer lightning flashes in other months. A similar trend was reported by [29] for the 2010–2014 period. To investigate the regional variations, Figure 2b shows the mean monthly NLDN/WWLLN ratios for all the data (denoted by "All") over the modeling domain and across the U.S. climate regions as shown in Figure 1. In Figure 2b, the monthly mean ratio is calculated as follows: first, the average of the monthly lightning flash rate for each region for all three years in that month for NLDN and WWLLN, respectively, is calculated, then the NLDN lightning flash rate is divided by the corresponding WWLLN value. Even though there were some regional variations, the seasonal variation patterns were similar across the U.S. climate regions with higher ratios during warm months and smaller ratios during cool months, except in the Northwest region where lightning activity is usually sparse.

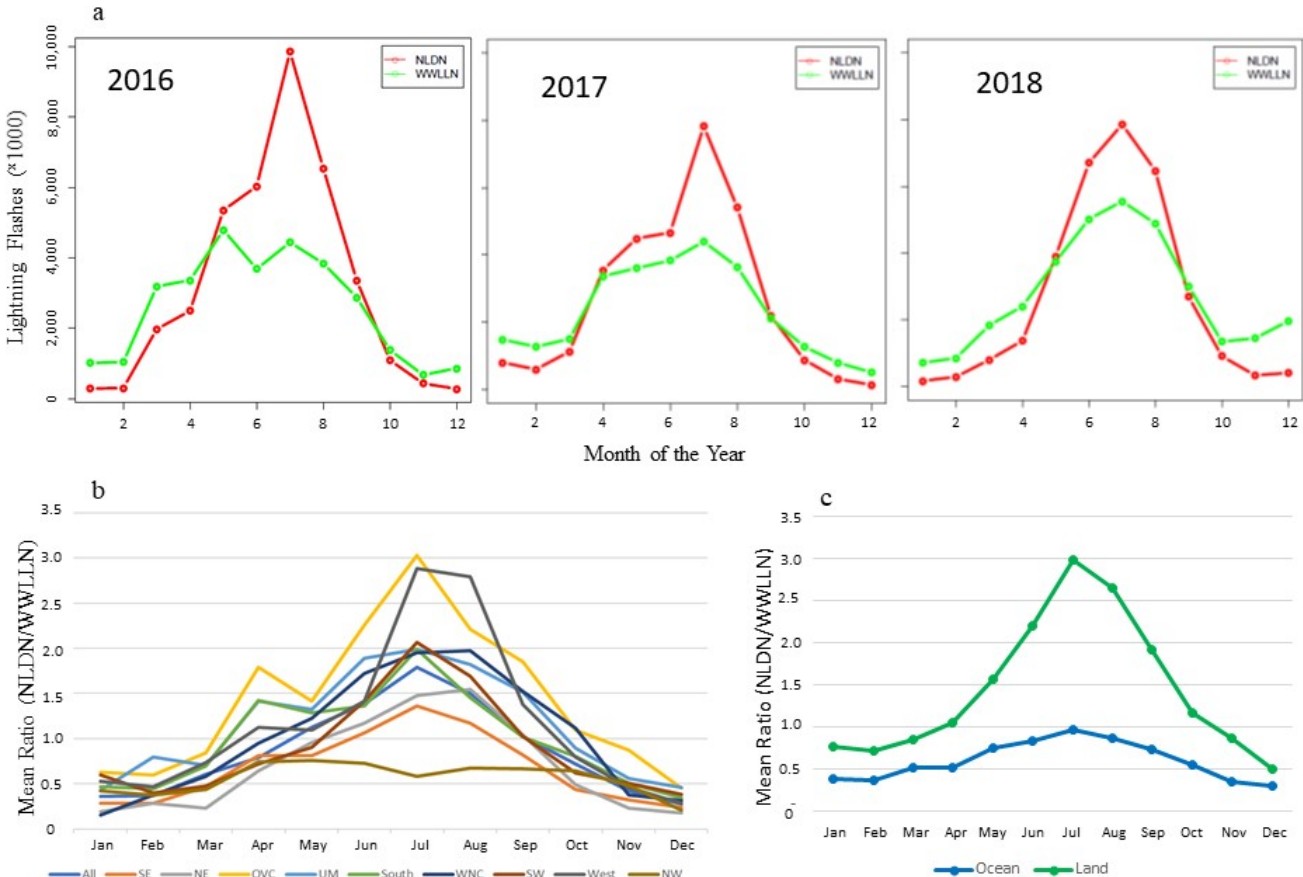

**Figure 2.** Lightning flashes from NLDN and WWLLN and the monthly ratios between NLDN and WWLLN over the CONUS and surrounding regions. (**a**) The total monthly lightning flashes detected by NLDN and WWLLN from 2016 through 2018, (**b**) The mean monthly ratios between NLDN and WWLLN flashes during 2016–2018 over CONUS and the U.S. climate regions, and (**c**) The mean monthly ratios between NLDN and WWLLN flashes during 2016–2018 over land and ocean.

### 3.2. The Scaling of the WWLLN Lightning Flash Rate to Generate $LNO_x$ Emissions

It would be ideal to scale the WWLLN lightning flashes using the monthly region specific NLDN/WWLLN ratios as shown in Figure 2b for applications over the CONUS using WWLLN data. However, the primary purpose in adopting the WWLLN data is for use in hemispheric and global applications for $LNO_x$ emissions, and it would be unrealistic

to use the regional specific scaling factors that are derived only from the U.S. climate regions. Since the mean ratios over the regions display similar seasonal trends, the $LNO_x$ emissions can be approximated using the WWLLN data for areas outside the U.S. by implementing a broader level of spatial aggregation by distinguishing between only the grid cells over land and over ocean to examine the seasonal trend of lightning flash ratios within the modeling domain. As indicated in Figure 2c and listed in Table 1, the lightning flash rate differences for the two networks are much larger over land than over ocean, especially during warm months. Figure 3 displays the original WWLLN, NLDN, and the adjusted (scaled) WWLLN (WWLLNs) lightning flash rate during July 2016. The lightning flash rates in WWLLNs are the original WWLLN flash rates multiplied by the NLDN/WWLLN ratios as shown in Figure 2c and Table 1, depending on the month of the year and the grid cell categorization (either over land or over ocean). Even though the spatial patterns are similar, the WWLLN reported much fewer lightning flashes than NLDN over land in this summer month as indicated by Figure 3a,b. After applying the scaling factors, the lightning flash rates in Figure 3c are more comparable to the lightning flash rates detected by NLDN (Figure 3b) in spatial distribution and intensity (similar results exist in other months). The NLDN network is devised specifically for the CONUS and its land-based monitors provide high detection efficiency for lightning activities occurring over the CONUS. Even though monitors are also stationed along the coastlines, the detection efficiency decreases as the distance of lightning activity from the coastline increases [39]. In practice, when WWLLN data are used to generate $LNO_x$ emissions in regional to global scale modeling, it is reasonable to apply the scaling factors listed in Table 1 only to grid cells over land and use the original WWLLN values for ocean grid cells (i.e., keep the scaling factor as 1 for grid cells over ocean). In this regional application, we use scale factors from Table 1 for both land and ocean.

**Table 1.** The monthly NLDN/WWLLN lightning flash ratio over land and ocean over the CONUS and surrounding regions.

| Month | Land | Ocean |
|---|---|---|
| January | 0.76 | 0.37 |
| February | 0.70 | 0.35 |
| March | 0.85 | 0.52 |
| April | 1.04 | 0.52 |
| May | 1.57 | 0.75 |
| June | 2.21 | 0.82 |
| July | 2.99 | 0.96 |
| August | 2.64 | 0.86 |
| September | 1.92 | 0.73 |
| October | 1.16 | 0.54 |
| November | 0.86 | 0.35 |
| December | 0.50 | 0.29 |

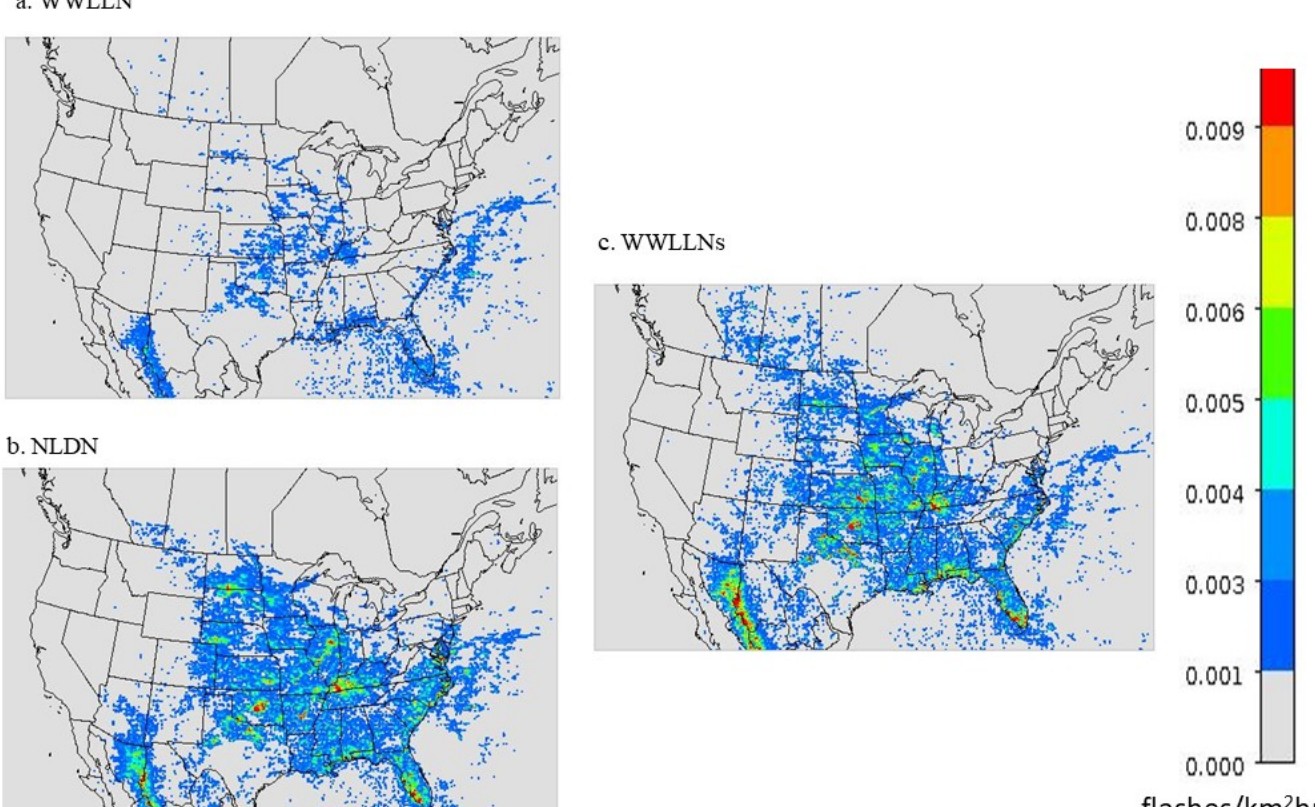

**Figure 3.** The original lightning flash rate detected by WWLLN (**a**) and NLDN (**b**), and the scaled WWLLN (WWLLNs) (**c**) lightning flash rate by applying the ratios as displayed in Figure 2c, depending on either land or ocean grid cell categorization during July 2016.

## 4. Results and Analysis

The impact of $LNO_x$ on air quality using NLDN data over the CONUS domain has been studied for different years [5,16]. In this study, our analysis is focused on assessing model performance when $LNO_x$ is generated using raw WWLLN data and WWLLN data scaled by NLDN data, though the analysis of the simulations using no $LNO_x$ and NLDN-based $LNO_x$ presented here can be used to further verify that the impact of $LNO_x$ on air quality for 2016 is consistent with previous studies for different years.

### 4.1. Surface Air Quality

The mean diurnal variations of hourly $O_3$ and timeseries of daily maximum 8-h $O_3$ (DM8O3) over the CONUS and selected climate regions (Figure 1) during July 2016 are provided in Figures 4 and 5. The regions presented are selected based on the impact of $LNO_x$ on air quality as discussed in [17], though the specific states included in the regions are slightly different than those in [17]. With the current model configuration and emissions inventory, inclusion of $LNO_x$ generally results in a larger overestimation of surface $O_3$ than the already overestimated $O_3$ in the model simulation without any $LNO_x$ (BASE) in the eastern U.S., particularly in Southeastern (SE) region. In the Southwest (SW) and West North Central (WNC) regions, the model performed better when $LNO_x$ was included by reducing the underestimation of $O_3$ present in the BASE simulations. Because the impact of $LNO_x$ on surface $NO_x$ levels has been previously shown to be generally negligible [17], the surface $NO_x$ mixing ratios are not shown here. Both the diurnal profiles and the time series of DM8O3 indicate the smaller impact on $LNO_x$ produced by the original WWLLN data (due to its low detection efficiency) than that produced by NLDN data. However, the simulation using the scaled WWLLNs data produced similar results to the simulation that

used NLDN. The lines representing WWLLNs, except for occasional lower values, closely follow the lines representing NLDN.

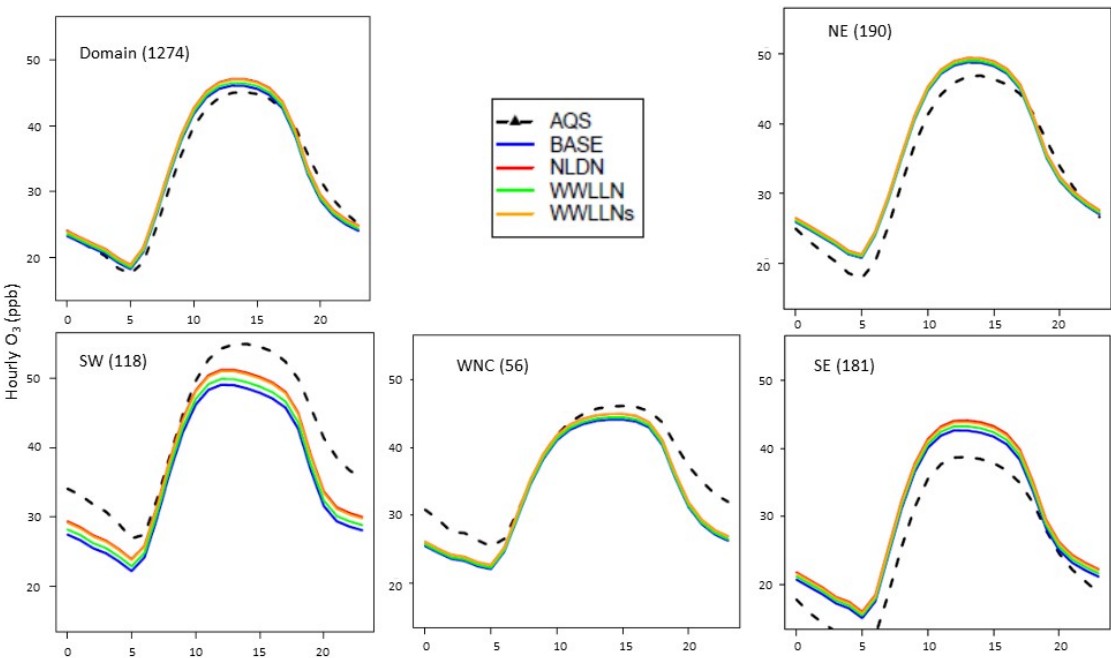

**Figure 4.** Mean diurnal variations of hourly O$_3$ mixing ratios in the domain and selected regions (NE, SE, WNC, and SW) during July 2016. The numbers in parentheses following the region names are the number of observation sites from AQS. NE = Northeast, SE = Southeast, WNC = West North Central, SW = Southwest (See Figure 1).

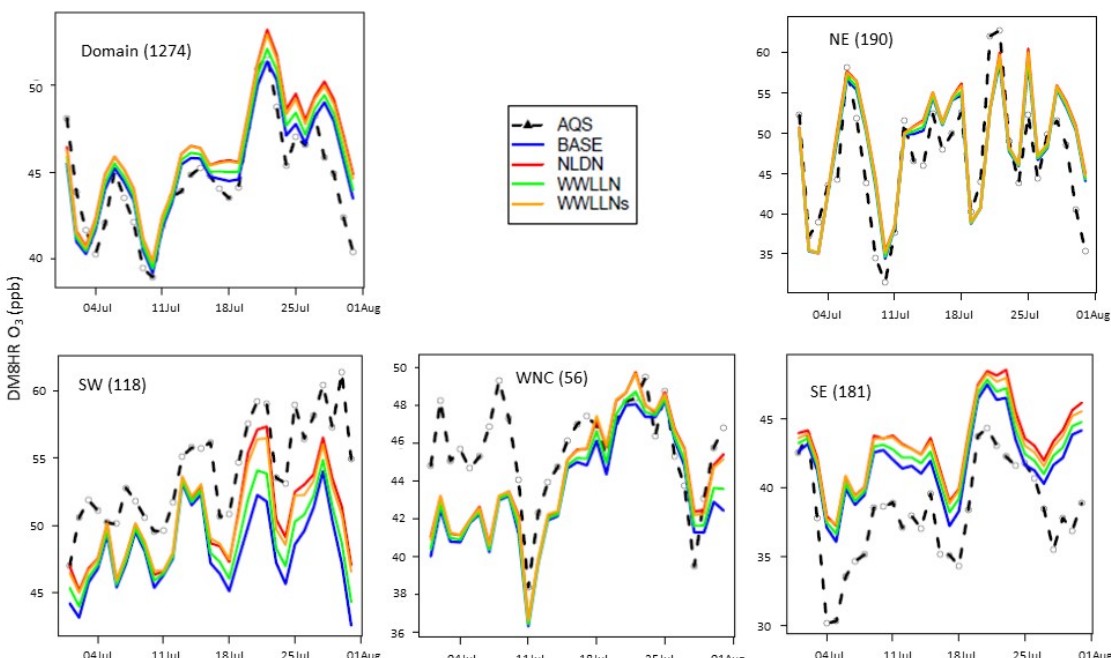

**Figure 5.** Timeseries of mean daily maximum 8-h O$_3$ mixing ratios in the domain and selected regions (NE, SE, WNC, and SW) during July 2016. The numbers in parentheses following the region names are the number of observation sites from AQS. NE = Northeast, SE = Southeast, WNC = West North Central, SW = Southwest (See Figure 1).

Figure 6 displays the mean RMSE differences of DM8O3 between the sensitivity cases and the BASE case during July 2016 at all the AQS sties over the CONUS to help highlight the varying effect of $LNO_x$ using different datasets over geographic regions. Similar to previous findings [5,17], the most significant impact of $LNO_x$ on surface $O_3$ occurred in the Southwest (SW) and WNC regions indicated by the large reduction in RMSE values (up to 3 ppb) in those regions, although noticeable impacts also occurred throughout the eastern U.S. with increases in RMSE values up to 2 ppb. Comparing amongst the three sensitivity cases, the WWLLN had the least impact on RMSE values, the NLDN had the largest impact, and the WWLLNs closely followed the NLDN in both spatial distribution and magnitude of impact on RMSE. In summary, the impact of $LNO_x$ estimated using the WWLLNs data on surface $O_3$ over the CONUS closely resembles that when $LNO_x$ is estimated using the NLDN data.

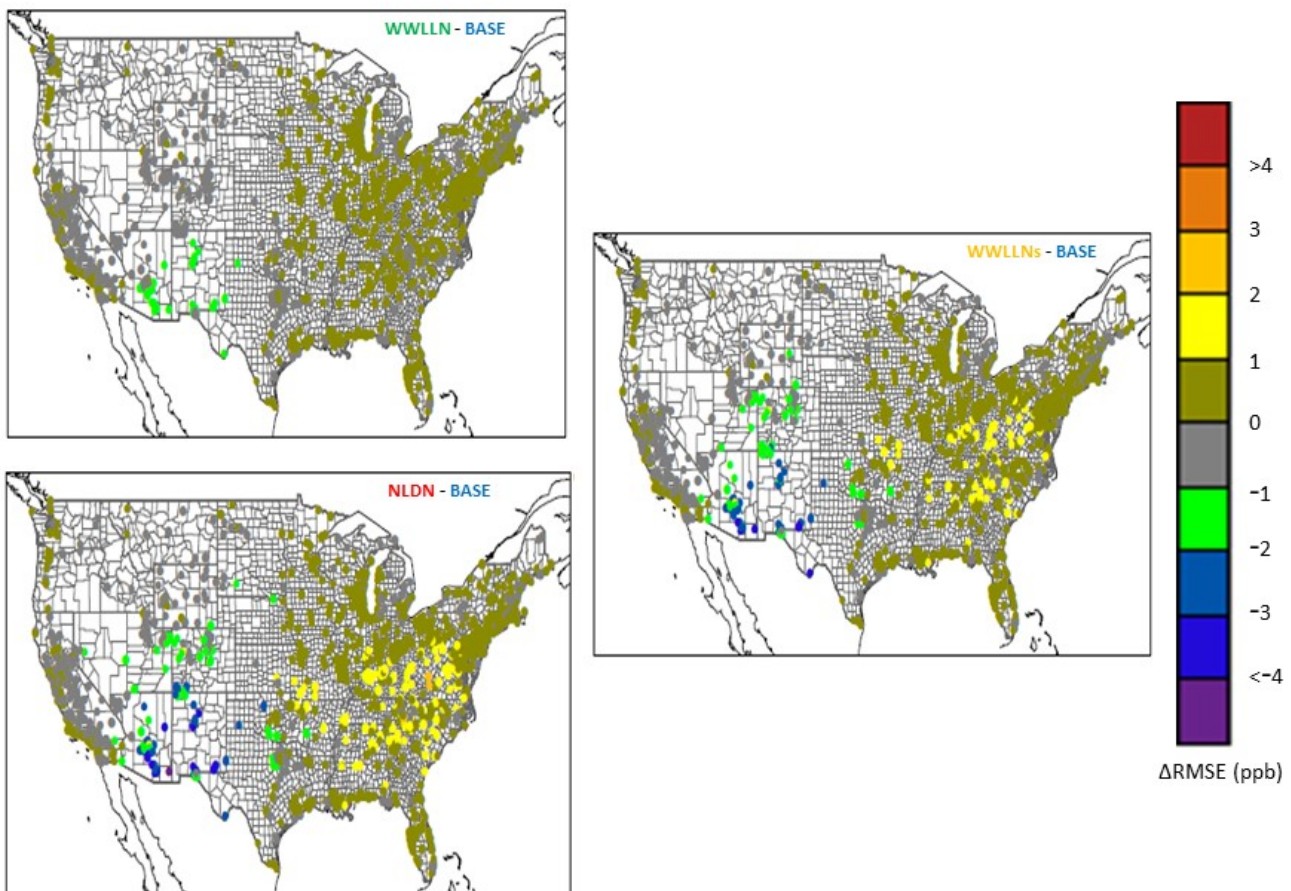

**Figure 6.** The change of RMSE for daily maximum 8-h $O_3$ at all the AQS sites in July 2016 for the sensitivity cases compared to the BASE case.

*4.2. Vertical Profiles*

4.2.1. Comparison with Ozonesonde Measurements

Since $LNO_x$ is primarily produced in the mid to upper troposphere, its direct and immediate impact on air quality also occurs in the mid to upper troposphere. Figure 7 displays $O_3$ vertical profiles as measured by ozonesondes at three locations (shown in Figure 1) and the corresponding time-space paired values simulated by all the model cases for July 2016 on a day (of 3 or 4 available days) when most $LNO_x$ impact was observed at the location. The model cases with $LNO_x$ produced more $O_3$ than the BASE simulation and brought the vertical profiles closer to the measured profiles at the Wallops and Boulder sites. Among the three sensitivity cases, WWLLN produced the least $O_3$ in the mid and upper troposphere, the NLDN produced the most $O_3$, and the WWLLNs produced slightly

less $O_3$ than the NLDN case, but again closely followed the vertical profile associated with the NLDN case demonstrating the effectiveness of scaling the WWLLN data in estimating $LNO_x$ emissions.

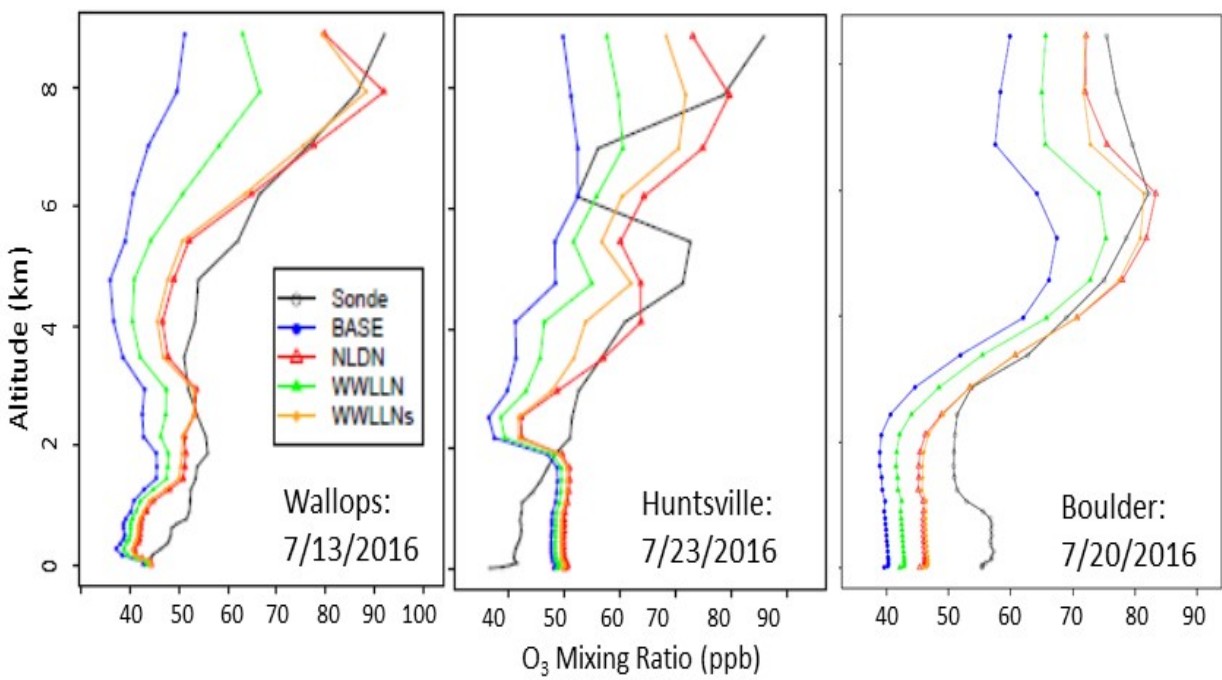

**Figure 7.** Comparison of simulated vertical $O_3$ profiles with ozonesonde measurement at three locations (Wallops, VA, USA, Huntsville, AL, USA, and Boulder, CO, USA) on days when lightning had significant impact and observations were available during July 2016.

### 4.2.2. Comparison with OMI $NO_2$ VCDs

Satellite-derived $NO_2$ VCDs provide the spatial distribution of the total tropospheric $NO_2$ densities that can be employed to assess the impact of $LNO_x$ on the modeled $NO_2$ columns across the modeling domain. Figure 8 displays the OMI retrieved and CMAQ simulated $NO_2$ VCDs, and the differences between simulated and observed values over the CONUS during July 2016. OMI retrievals were recalculated using the simulated CMAQ vertical $NO_2$ profile from each $LNO_x$ emissions case. Compared to OMI retrievals, the BASE model simulation shows a significant underestimation of $NO_2$ VCDs across space as reflected in the difference plot. The NLDN case generated larger $NO_2$ VCD values than the BASE case, especially in the east-central regions where lightning activities are prevalent during summer months. The difference plot indicates that the NLDN case generally reduced the biases across space with slight overestimation in the central areas compared to the BASE case. Similar to the BASE case, the WWLLN case underestimated $NO_2$ VCDs across the domain with negligible increases in $NO_2$ values. Although VCDs from the WWLLNs case are lower than for the NLDN case, the WWLLNs case generally follows the pattern of the NLDN case in space. Note that $LNO_x$ is only a portion in the total $NO_x$ budget, and its percentage of the total $NO_x$ is much lower in urban areas than in rural areas. Therefore, the comparisons should only be interpreted in a relative sense (because the $NO_2$ VCDs represent total $NO_2$) when interpreting the different approaches to represent $LNO_x$. However, the analysis further demonstrates that the scaled WWLLN data can produce $LNO_x$ value that are comparable in time and space with the NLDN data.

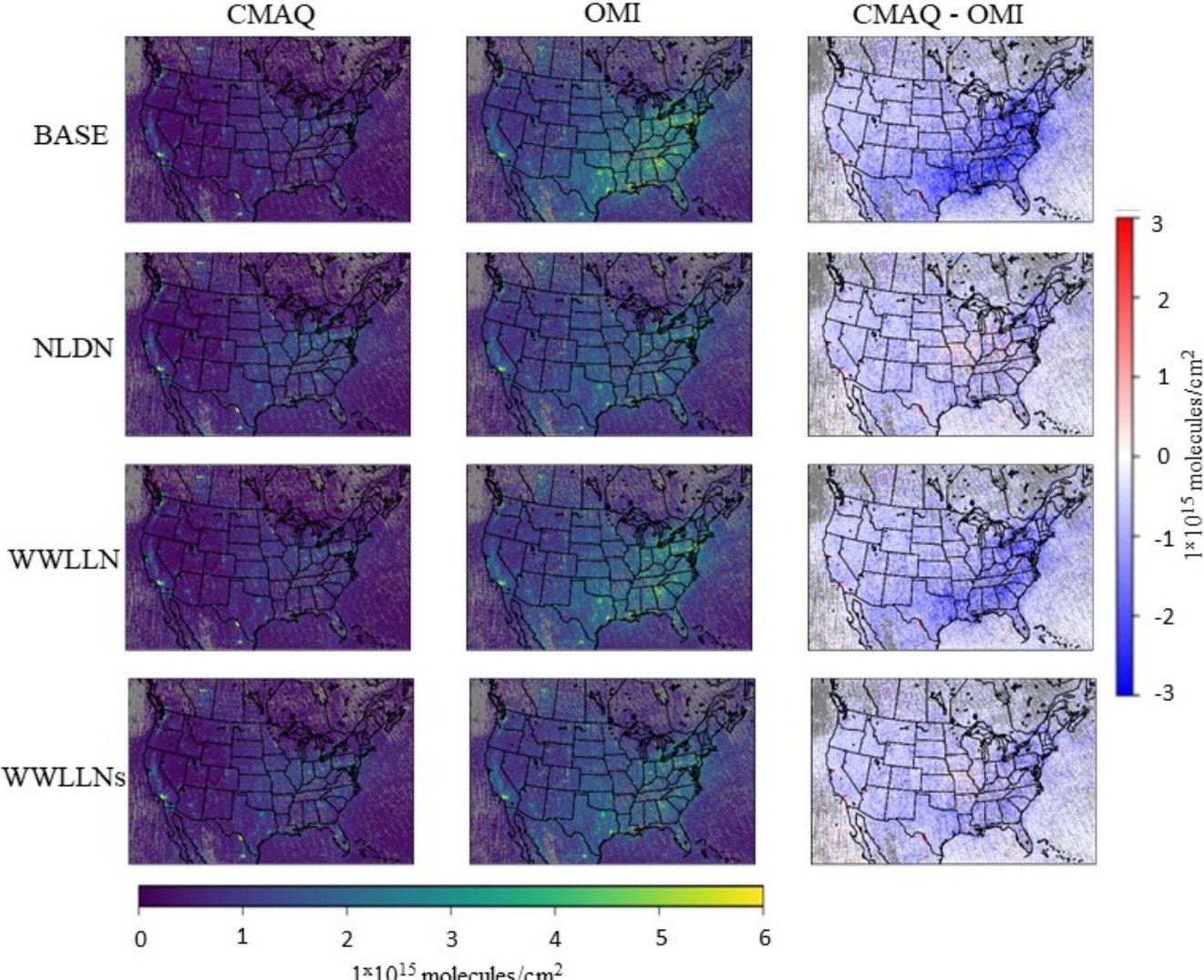

**Figure 8.** The NO$_2$ VCDs (averaged over July 2016) retrieved from OMI and simulated by CMAQ over the CONUS. The left column is CMAQ simulations, the middle column is OMI retrievals by applying the corresponding air mass factor (AMF) using CMAQ simulated NO$_2$ mixing ratios (observed NO$_2$ mixing ratio profiles), and the right column displays the difference between simulated and OMI observed NO$_2$ VCDs. The rows from top to bottom correspond to the four model cases: BASE, NLDN, WWLLN and WWLLNs. The legend bar at the bottom applies to OMI and simulated NO$_2$ VCDs (**the left and middle columns**), and the legend bar on the right applies to the difference plots (**the right column**).

*4.3. Wet and Dry Nitrate Deposition*

The modulation of the tropospheric loading of oxidized nitrogen by LNO$_x$ emissions also impacts the magnitude of atmospheric deposition of oxidized nitrogen to sensitive ecosystems. Figure 9 displays the monthly mean observed and simulated wet NO$_3^-$ deposition and the statistical metrics across 232 NADP/NTN measurement sites in the CONUS during 2016. The accuracy of simulated wet deposition is determined by the accuracy of the predicted precipitation and the simulated ambient concentrations of a variety of oxidized nitrogen species (eventually represented by NO$_3^-$ concentrations in rainwater). Since all the model cases have the same meteorological fields, the differences among model cases are solely determined by the simulated ambient oxidized nitrogen concentrations. As shown in Figure 9a, the wet NO$_3^-$ deposition exhibits higher values during the warm months than in cool months in response to convective conditions that

often lead to precipitation and lightning. With a few exceptions, all the model cases generally tended to underestimate the wet $NO_3^-$ deposition throughout the year, and the underestimation peaked in July. The BASE model significantly underestimated the wet deposition during summer months by up to 30%, with the largest bias occurring in July (Figure 9d,e). Corresponding to the $LNO_x$ emission increases from WWLLN to WWLLNs to NLDN, the simulated wet $NO_3^-$ deposition also increases, with the NLDN simulations almost matching the observed deposition levels. Though the differences in RMSE and correlation coefficients among the model cases were small (likely due to uncertainties in the precipitation fields common to all simulations), the model cases with more $LNO_x$ emissions changed model performance results in the right direction. In summary, the WWLLNs case closely follows the NLDN case in the monthly mean wet deposition and all the associated model performance statistics.

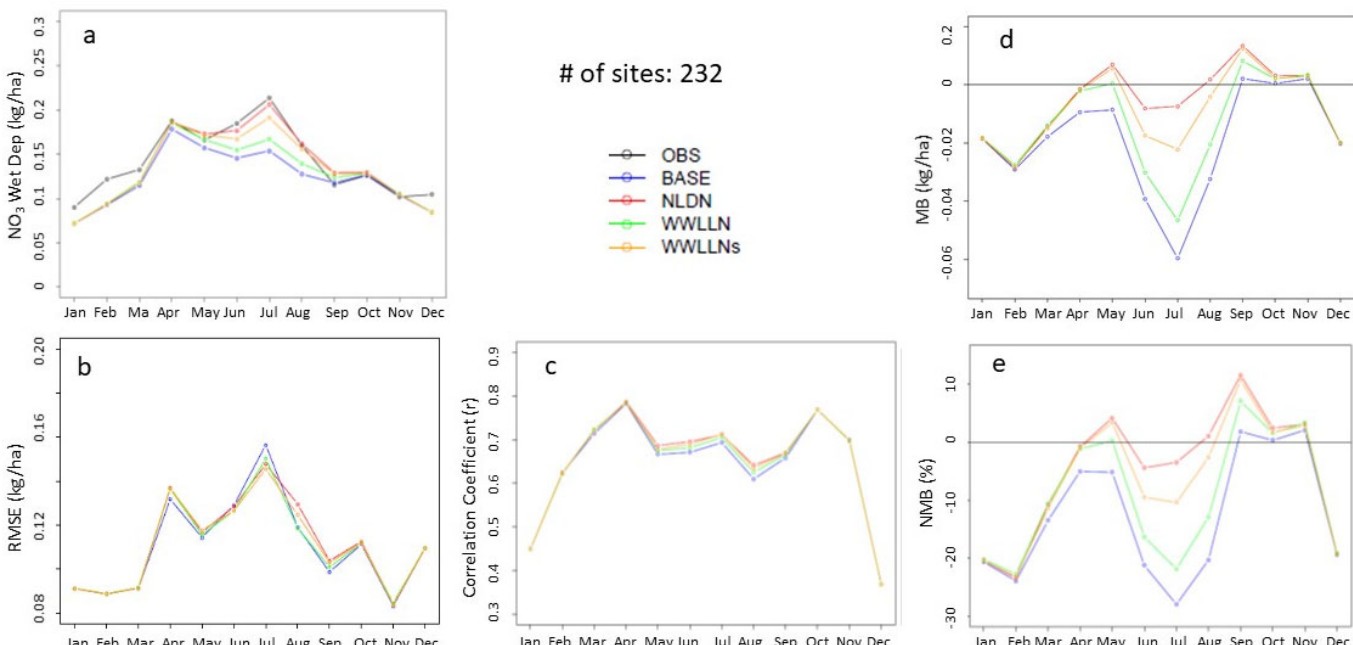

**Figure 9.** The monthly mean observed and simulated wet $NO_3^-$ deposition and statistic metrics over CONUS in 2016: (**a**) Monthly mean deposition at 257 NADP/NTN sites, (**b**) RMSE, (**c**) Correlation coefficient, (**d**) MB, and (**e**) NMB.

To examine the spatial distributions of wet $NO_3^-$ deposition, Figure 10 presents the monthly simulated values from the BASE case and the differences between the sensitivity cases and the BASE case for July 2016. The spatial patterns of wet deposition are the result of precipitation and oxidized-nitrogen species ($NO_y$) ambient concentrations, and during summer months, precipitation- and lightning-induced $NO_y$ ambient concentrations are highly correlated in time and space. Whereas the BASE map (Figure 10a) shows the wet $NO_3^-$ deposition attributed to non-lightning sources, the difference maps between the BASE case and the sensitivity cases indicate the incremental amount of wet $NO_3^-$ deposition resulting from $LNO_x$ emissions. As shown in Figure 10b–d, the increases in wet $NO_3^-$ deposition were the direct result of the lightning activities (Figure 3). Due to lower detection efficiency in the original WWLLN data (and consequently lower $LNO_x$ and $NO_y$), the increase in wet $NO_3^-$ deposition is the least (Figure 10b). The NLDN case produced the largest increase (~24% averaged over the domain and up to more than a factor of 2 at certain individual grid cells) and the WWLLNs case followed closely. To further illustrate the impact of the $LNO_x$ emissions from the sensitivity cases on the wet $NO_3^-$ deposition at the NADP/NTN monitoring sites, Figure 11 presents the BASE case NMB values and the differences of NMB values between the sensitivity cases and the BASE case. The difference

plots (the difference between the absolute NMB values of a sensitivity case and of the BASE case) demonstrate whether the model performance is improved or degraded at the monitoring locations by the sensitivity cases. As shown in Figure 11a, the BASE model generally underestimated the wet $NO_3^-$ deposition at a majority of the monitoring sites with overestimation at sporadic locations (though it seems more systematic in the northwest corner of the domain). The addition of $LNO_x$ emissions in all the sensitivity simulations improved the model performance for simulated wet $NO_3^-$ deposition as indicated in Figure 11b–d by the reduced absolute NMB values at most of the monitoring sites (about three quarters of the 232 NADP/NTN sites showed reductions in absolute NMB values). In agreement with Figure 10, all the sensitivity cases produced the largest reductions at many sites in the central and southeastern regions and the fewest reductions in the northwest regions. The NLDN case performed best with the largest reductions (though the number of sites that have reduced NMB values is slightly smaller than the other cases due to small overestimations at certain sites by the NLDN case) followed closely by the WWLLNs case, and the WWLLN case produced the least improvement.

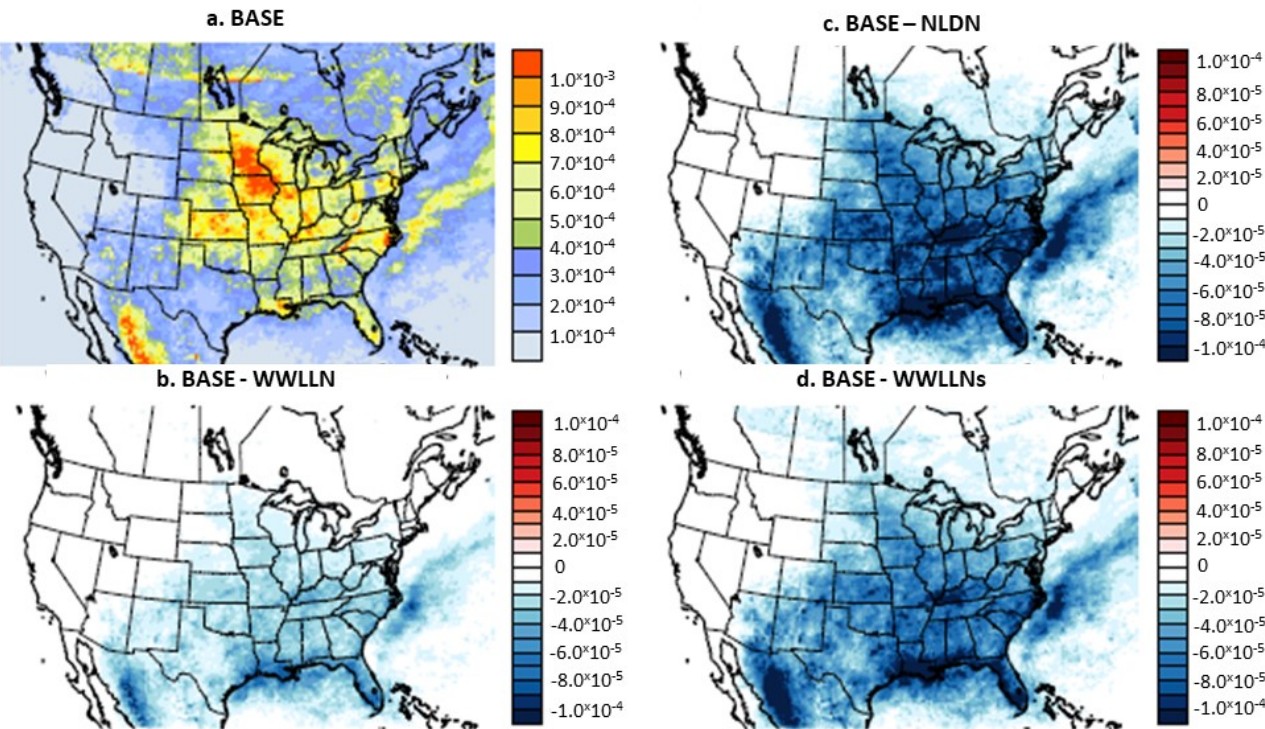

**Figure 10.** Monthly mean $NO_3^-$ wet deposition simulated by BASE (**a**) and the differences between sensitivity cases and the BASE (**b–d**) during July 2016. Unit: kg/ha.

Unlike wet $NO_3^-$ deposition, dry oxidized-nitrogen deposition at a particular location is primarily driven by the ambient $NO_y$ concentrations and characteristics of the underlying earth surface. As demonstrated in Figure 12a, large dry oxidized-nitrogen deposition tended to occur over more populated regions where the ambient $NO_y$ concentrations were primarily attributed to anthropogenic sources. The $LNO_x$ emissions contributed varying fractions (~5% over the domain and up to 30–50% at some individual grid cells when the NLDN case was compared with the BASE case in July 2016) to the monthly mean dry oxidized-nitrogen deposition in regions which experienced more frequent lightning activities. The model case with the original WWLLN data (Figure 12b) produced negligible changes, whereas the NLDN case (Figure 12c) and the WWLLNs case (Figure 12d) generated a similar effect in the increased amount of dry oxidized-nitrogen deposition spatially, further demonstrating that the scaled WWLLN data can be substituted for the NLDN data for $LNO_x$ emissions to produce similar results in various model applications.

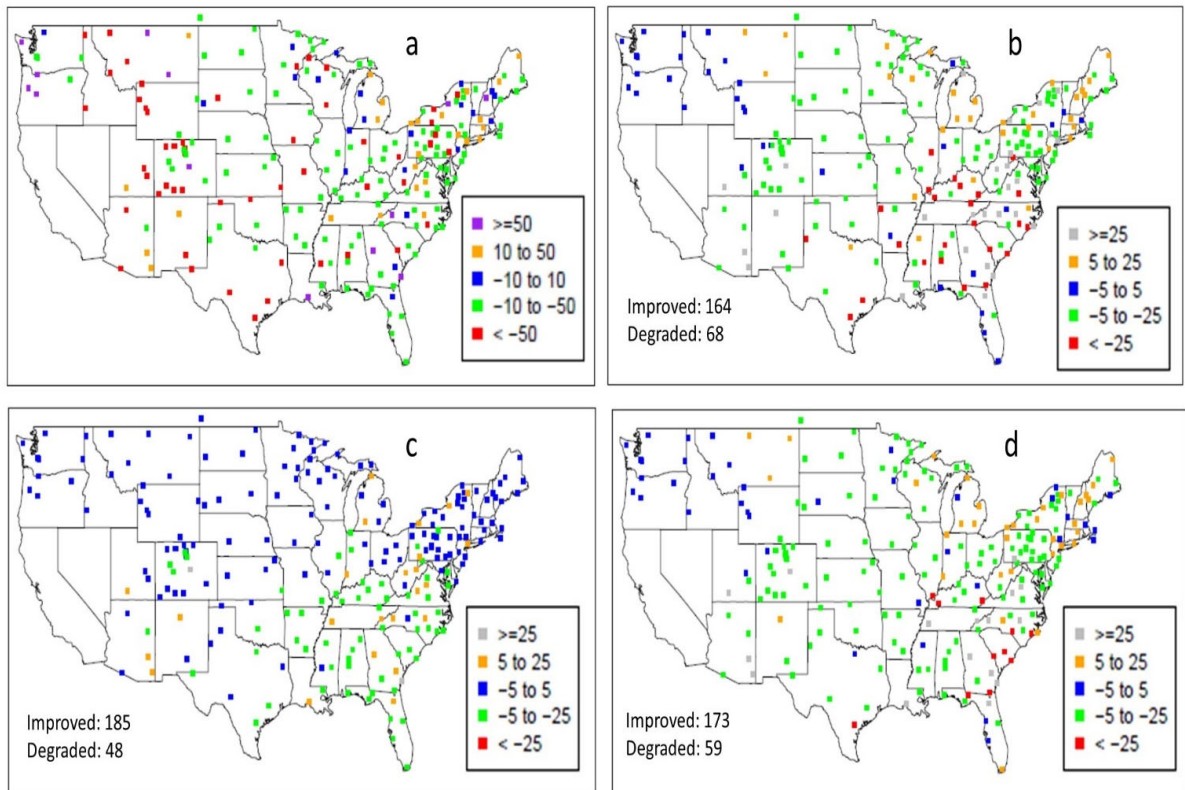

**Figure 11.** The NMB (%) values associated with wet $NO_3^-$ deposition simulated by BASE and the NMB differences between sensitivity cases and the BASE at NADP/NTN monitoring sites during July 2016: (**a**) BASE NMBs, (**b**) |NLDN NMB| − |BASE NMB|, (**c**) |WWLLN NMB| − |BASE NMB|, and (**d**) |WWLLNs NMB| − |BASE NMB|. The numbers after "Improved" and "Degraded" marked on the difference plots indicate the number of sites that have reduced biases (improved) and the increased biases (degraded) by the sensitivity cases, respectively.

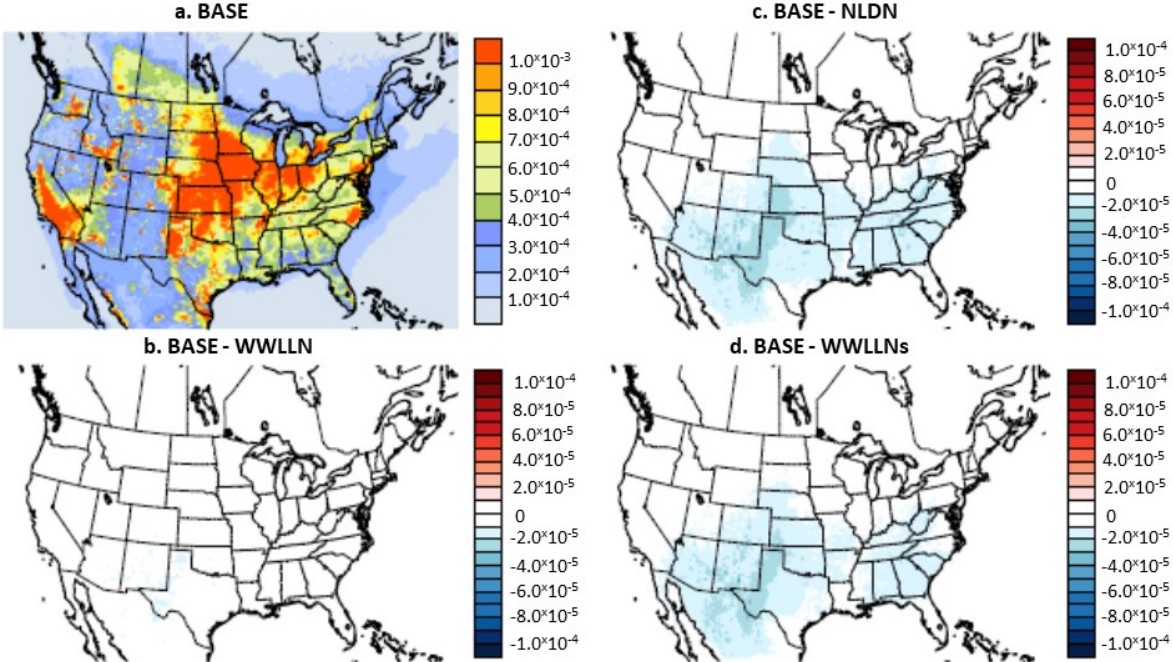

**Figure 12.** Monthly mean $NO_3^-$ dry deposition simulated by BASE (**a**) and the differences between sensitivity cases and the BASE (**b**–**d**) during July 2016. Unit: kg/ha.

## 5. Conclusions and Remarks

Lightning-induced NO$_x$ emissions have previously shown profound impacts on atmospheric NO$_x$, O$_3$, and NO$_3^-$ concentrations and the wet and dry NO$_3^-$ deposition at the surface. Inclusion of lightning flash data observed from ground-based networks can significantly enhance the accuracy of model simulated LNO$_x$ emissions in time and space, improving model performance against a variety of observations. In this study, three years of lightning flash data from two ground-based lightning detection networks, NLDN and WWLLN, with varying detection efficiency and spatial coverage were analyzed. Comparison of lightning flash rates from both networks over the CONUS indicates that the ratio of detected lightning flashes displays consistent seasonal variations despite slight regional differences over the U.S. climatological regions. Under the assumption that NLDN detected lightning flashes represent the ground truth, we scaled the WWLLN data by the monthly mean ratios between NLDN and WWLLN flashes grouped into either land-based or ocean-based grid cells. CMAQ simulations configured with the different lightning flash data resulting in different LNO$_x$ emissions revealed that the scaled WWLLN data led to improved model performance over the original WWLLN data and comparable results with the NLDN data in all the aspects assessed in this study.

Consistent with previous work [17], the impact of LNO$_x$ emissions on model performance in predicting surface O$_3$ mixing ratios varies in both spatial distribution and magnitude, but the simulated O$_3$ vertical profiles and NO$_2$ VCDs all improved incrementally with the inclusion of LNO$_x$, corresponding to the increased accuracy of LNO$_x$ emissions as indicated by ozonesonde and satellite observations. Taking advantage of annual simulations with different LNO$_x$ configurations, we analyzed the impact of LNO$_x$ emissions on wet and dry atmospheric deposition of oxidized nitrogen across the CONUS. During summer months, the CMAQ model BASE case underestimated the monthly wet oxidized nitrogen deposition by up to 30% without LNO$_x$ emissions averaged over all the NADP/NTN sites across the CONUS, which still signifies a large improvement over the BASE model compared to past similar studies using earlier versions of the same modeling system [24]. The underestimation was ameliorated gradually by increasing the LNO$_x$ emissions from WWLLN to WWLLNs and then to NLDN, suggesting the important role that LNO$_x$ emissions play in the formation and deposition of atmospheric oxidized nitrogen. In response to the spatial pattern of lightning activities, the improvement of model performance displayed distinct spatial variations. By taking the difference between the NLDN case and the BASE case, it is estimated that LNO$_x$ emissions contributed ~24% to wet oxidized nitrogen deposition and ~5% to dry oxidized nitrogen deposition averaged over the modeling domain during July 2016. At certain grid cells located in remote areas with low anthropogenic NO$_x$ emissions, the LNO$_x$ emissions increased wet oxidized nitrogen deposition by more than a factor of 2 and dry oxidized nitrogen deposition by over 50%, respectively.

Owing to many years' effort in field measurements and modeling studies, our understanding of LNO$_x$ emissions and its important roles in various atmospheric processes has been greatly improved. The ground-based lightning detection networks are invaluable in providing continuous lightning flash data across large geographic regions. However, due to complexity in detection and data retrieval technologies, each network has its unique limitations in terms of detection efficiency, spatial coverage, and the cost to acquire and use the data. For application of LNO$_x$ emissions over the CONUS, the NLDN data is the best option, as it provides one of the most accurate lightning flash datasets over the CONUS (and is often treated as ground truth); however, its use does incur a significant cost. For applications outside the CONUS domain, the WWLLN data serves as a good alternative, though its detection efficiency is lower than the NLDN data and acquiring it may also include a small cost under some conditions. In this work, we adapted the WWLLN data based on the monthly mean ratios between NLDN and WWLLN to produce LNO$_x$ emissions in the CMAQ model and demonstrated that the model case with the scaled WWLLN data achieved similar performance in all the aspects in terms of LNO$_x$ as the model simulation

using NLDN data. One caveat is that the scaling factors were derived using data from 2016–2018 over the CONUS domain, when these factors are applied to model simulations outside the CONUS domain and for other years, caution should be exercised in interpreting model results. Nevertheless, this is the best approximation to be made based on the data availability and modeling assessment. These scaling factors have also been applied to Northern Hemispheric modeling studies and the simulation results are currently being analyzed and will be presented in future work. This ongoing research is aimed at advancing methods for considering and using the lightning observations that have become available to provide $LNO_x$ emissions in regional air quality modeling. As lightning observations from other regional and global networks become available, a composite of lightning data from different networks could be employed to improve the accuracy of $LNO_x$ estimates across various modeling scales. With the advancement of lightning detection techniques, more detailed properties associated with the process of lightning discharge (such as the lightning energy level and the separation of cloud-to-ground and inter- or intra-cloud strikes) are being more accurately quantified, especially with the available satellite lightning products from Geostationary Lightning Mapper (GLM) detection systems borne on the GOES-16 and -17 satellites [40]. Therefore, we expect the lightning $NO_x$ production schemes in air quality models to continue to evolve, building upon the methodology presented here.

**Author Contributions:** Conceptualization, D.K.; Formal analysis, D.K., C.H. and J.D.E.; Methodology, D.K. and G.S.; Project administration, D.K.; Supervision, D.K.; Writing—original draft, D.K.; Writing—review and editing, D.K., C.H., G.S., J.D.E., J.M.M., R.M. and B.H.H. All authors have read and agreed to the published version of the manuscript.

**Funding:** This research received no external funding.

**Institutional Review Board Statement:** Not applicable.

**Informed Consent Statement:** Not applicable.

**Data Availability Statement:** The raw lightning flash observation data can be purchased through Vaisala Inc. (https://www.vaisala.com/en/products/systems/lightning-detection (accessed on 4 August 2022)), and the WWLLN raw data are also available for purchase at http://wwlln.net (accessed on 4 August 2022). All other data is available upon request.

**Acknowledgments:** The authors thank Wyat Appel and Jeff Willison for technical reviews of the initial draft of this manuscript. James D. East and J. Mike Madden are supported in part by an appointment to the ORISE participant research program supported by an interagency agreement between EPA and DOE.

**Conflicts of Interest:** The raw lightning flash observation data can be purchased through Vaisala Inc. (https://www.vaisala.com/en/products/systems/lightning-detection (accessed on 4 August 2022)), and the WWLLN raw data are also available for purchase at http://wwlln.net (accessed on 4 August 2022). All other data is available upon request.

**Disclaimer:** This paper has been subjected to an EPA review and approved for publication. The views expressed in this paper are those of the authors and do not necessarily represent the views or policies of the U.S. EPA.

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
