# Peer review of "Assessing the Impact of Lightning NOx Emissions in CMAQ Using Lightning Flash Data from WWLLN over the Contiguous United States"

_atmosphere, doi:10.3390/atmos13081248_

Round 1

Reviewer 1 Report

This review is about the article “Assessing the Impact of Lightning NOx Emissions in CMAQ Using Lightning Flash Data from WWLLN over the Contiguous United States”; by Daiwen Kang et al.

In general, I found the manuscript very informative, clear, well written, and easy to follow. All sections are well written and mapped with the title. However, the following minor points may be incorporated before publishing.

Lines 13-38: The abstract is too long and not properly focused on the scope and respective outcome of the study. I suggest having a concise version of the current abstract and align properly to reflect the scope and outcome of the study.

Lines 101-102: Authors should explain what is/are the rationale behind the selected period (2016-2018) for Lightning flash data.

Line 249: I suggest renaming Section 4: “Results” as “Results and Discussion”.***

Line 434: I suggest renaming Section 5: “Conclusions and Remarks” as “Conclusions”. Most of the parts in Section 5 are not suitable to present under “Conclusions”. Some of the parts (Lines: 444-450, 451-457, 473-494) should be moved to Section 4*** and realign the text accordingly.

Figure resolutions and quality are very low henceforth enhancing the readability of the figures is a must before publishing. Legends of Figures 4, 5, 7, & 9 are too large.

It is not clear why the authors considered the flash density (rate as authors stated) on an hourly base instead of daily/monthly. Ex: Figure 3 (representing July 2016 observations).

How did you calculate the lightning flash ratio over land and ocean (ex: Table 1)? Is it the ratio of flash rate or flash count? Better to take the ratio of flash density/rate.

Reviewer 2 Report

This manuscript describes how WWLLN lightning flashes can be used in regional or hemispheric CMAQ modeling efforts after scaling the WWLLN data using network (such as NLDN) data with much better detection efficiency.  The results are clearly demonstrated in the paper in the form of maps showing improvement of the model for surface O3, O3 profile, NO2 tropospheric column, and wet and dry deposition.  A few minor edits are required before publication as outlined below.

line 27-28:  it is almost not worth showing the model results using the raw WWLLN data because its detection efficiency is so poor over land.  However, it would take a lot of work to remove the plots associated with the simulation using these data.

line 58:  add Allen et al. (2012) reference here.  Suggested text is:

....became readily available a parameterization scheme for CMAQ model flash rates was derived such that predicted monthly flashes were scaled to the NLDN observed flashes (Allen et al., 2012).  This scheme was improved such that inline LNOx production was based on hourly observed lightning flashes and implemented in....

lines 77-80:  With LNOx included didn't wet NO3 deposition nearly match the observed in Allen et al. (2012)?

line 170:  I think NLDN uses 400 Hz to 400 kHz

line 183:  but most of what WWLLN detects is CG

line 337:  total NO2 instead of total NOx

line 343:  observed NO2 mixing ratio profiles

lines 360-361:  not true for NLDN in May and Sept - Nov

line 376:  change depositions to deposition throughout the paper

line 455:  Taking advantage of...

line 458:  CMAQ model BASE case

line 480:  best option for CG flashes; Earth Networks might not agree that NLDN is the most accurate.   Maybe change this statement to say that NLDN is one of the most accurate flash data sets over CONUS

Reviewer 3 Report

This paper is a meaningful practice of estimating lightning induced NOX emissions by comparing NLDN and WWLLN lightning flash data. The comparable performance between two data sets suggests that scaled WWLLN may be used as a substitute for NLDN data in air quality models. The paper is well written and the text is clear and easy to read. The conclusions are consistent with the evidence and arguments presented.  The scientific topic discussed in this article is a current and relevant academic topic. The structure of the article is appropriate and presents good quality, while linking the results to the text. Conclusion is conveniently linked to the object that has been proposed as a research question, in order to apply the theory to a consistent work of scientific research.

I have only one question. The graphics resolution is too low to see clearly. Need to modify the pictures.

I think this manuscript can be accepted for published in this journal.

Author Response

We thank the reviewer for the overall positive and supportive comments.

The loss of figure resolution is probably the result of conversion and insertion of the figures during the editing process. We believe that we will have the opportunity to provide separate figures with high resolution during the final production stage.

Round 2

Reviewer 1 Report

The paper can be accepted in its current form, as the authors have made necessary changes to the manuscript according to the reviewers' comments. However, as the authors suggested, high-quality figures should be incorporated before publishing. Responses for the queries are also acceptable.